# MoS$^2$: Mixture of Scale and Shift Experts
# for Text-Only Video Captioning

## ABSTRACT

Video captioning is a challenging task and typically requires video-text paired data for training. However, manually annotating coherent textual descriptions for videos is laborious and time-consuming. To address this problem, we propose to utilize solely text data to enhance video captioning models. Drawing inspiration from the exceptional text generation capabilities demonstrated by large language models (LLMs), we aim to leverage these models to generate high-quality and high-diversity video captions for the target domain. Specifically, we prompt GPT-4 with few-shot target-domain captions to generate a limited set of plausible video captions. Subsequently, we continue to prompt GPT-4 with the generated captions to acquire large-scale captions. To fully exploit the generated captions, we propose a Mixture of Scale and Shift experts (MoS$^2$) for efficient adaptation of pre-trained image captioning models for video captioning. MoS$^2$ estimates a probability distribution over a collection of experts by a lightweight routing network, determining the allocation of tokens to appropriate experts. This dynamic adjustment mechanism allows for specific responses to input features, thereby enhancing the model's ability to handle data variations. Our approach not only customizes model responses to input variations, effectively addressing the distribution shift between synthetic and actual captions but also significantly reduces the number of learnable parameters, allowing for more efficient adaptations. With only text data, we achieve superior performance and significantly narrow the performance gap between zero-shot and fine-tuned models. Our method boosts video captioning performance with the synthetic text data, thus substantially alleviating the dependence on paired and large-scale real data of the target domain. The code will be publicly available.

## CCS CONCEPTS

• **Computing methodologies** → **Computer vision tasks**; **Natural language generation**.

## KEYWORDS

Video Captioning, Large Language Models, Mixture of Experts

## 1 INTRODUCTION

The objective of visual captioning is to generate coherent descriptions for visual content automatically. Recent studies [2, 7, 22, 23,

ACM MM, 2024, Melbourne, Australia
© 2024 Copyright held by the owner/author(s). Publication rights licensed to ACM.
ACM ISBN 978-x-xxxx-xxxx-x/YY/MM
https://doi.org/10.1145/nnnnnnn.nnnnnnn

32, 46, 47, 50, 60] have demonstrated that large-scale pre-training can significantly enhance zero-shot captioning performance. Despite notable advancements in zero-shot captioning methods [22, 24, 46], the performance of zero-shot captioning methods is largely lower compared to models [6, 13, 43, 48] fine-tuned with target-domain data. However, adapting vision-language models for each upcoming new domain is cumbersome, which requires fine-tuning the models with domain-specific datasets containing visual-text pairs. Besides expensive computational costs for models fine-tuning, building large-scale training datasets for captioning is also time-consuming and labor-intensive.

To improve the captioning performance on the target domain with lower training costs, some recent researches [11, 24, 34, 52] attempt to fine-tune the text decoder of the vision-language models using only textual data. DeCap [24] first projects visual and text embeddings into the CLIP space [35] and optimizes only the text decoder on a dedicated text corpus for image captioning. Subsequent studies [11, 34, 52, 57] further enhance the efficacy of text-only training by employing contrastive learning based models [35]. Current text-only training captioning works only focus on image captioning. These works extract textual training data from web-scraped image-text pairs or corpora tailored for specific target domains, potentially limiting the effectiveness and broad applications. Although only textual descriptions (*i.e.*, video captions) are required for training, we believe extracting high-quality text data for video captioning is even more challenging.

To cope with the issue of data collection, this work investigates video captioning from a different perspective. Different from prior works [6, 13, 28, 43, 48] fine-tune models with large-scale video-text pairs of the target domain, we aim to improve the video captioning performance on the target domain with only a few (*e.g.*, 10-shot) textual training examples of the target domain. Inspired by prior image captioning works [62], we try to generate a domain-specific text-only corpus to achieve text-only training. Since LLMs (*e.g.*, GPT-4 [1]) have superior instruction-following and contextual reasoning capabilities, we design few-shot instruction-following prompts request GPT-4 [1] to generate captions that closely align with the target domain. Compared to crawling text descriptions from the web [10], there is no need to manually craft complex filtering rules. In addition, the captions generated by LLMs are more similar to human-like styles, of higher-quality, free from grammatical errors and meaningless symbols. LLMs offer precise control over the text generation process through prompt engineering, which facilitates adherence to specific guidelines or themes.

Currrent pre-trained captioning models [2, 54 **?** ] typically incorporate large vision encoders [9, 35] and employ LLMs [58, 59] as the text decoders. Although only the text decoder is fine-tuned in

our text-only video captioning framework, optimizing all parameters within a large-scale text decoder remains costly. To further improve the training efficiency and preserve the foundational knowledge learned from the large-scale pre-training, we further explore parameter-efficient transfer learning, where a minimal set of parameters within the text decoder is updated.

Existing parameter-efficient fine-tuning techniques, though effective, typically assume minor differences between training and target domains. However, our approach, which prompts LLMs to generate domain-specific text, still faces challenges stemming from the distribution shift between synthetic and actual captions. To counter this, we develop a novel architecture, the Mixture of Scale and Shift Experts (MoS$^2$), which incorporates input-dependent adaptability into the model's parameters. A lightweight routing network estimates a probability distribution over a collection of experts, determining the allocation of tokens to appropriate experts. This dynamic adjustment mechanism allows for specific responses to input features, thereby enhancing the model's ability to handle data variations. Unlike traditional Mixture-of-Experts (MoE) networks that employ dense feed-forward networks with a high parameter count, our method simplifies the expert components to scale or shift factors, which significantly reduces the number of parameters, aligning with our goal of efficient training. Consequently, adaptive scale and shift factors are applied to individual tokens, enhancing flexibility and overall performance. By employing MoS$^2$, our approach not only customizes model responses to input variations, effectively addressing the distribution shift between synthetic and actual captions but also significantly reduces the number of learnable parameters, allowing for more efficient adaptations. Following the fine-tuning of the text decoder, we integrate it with the associate vision encoder for enhanced video captioning.

This paper presents a parameter-efficient text-only video captioning method trained with synthetic target-domain data. Extensive experiments compared to state-of-the-art methods validate the superiority of our proposed text-only video captioning method. The key contributions of this paper are summarized as follows:

- We conduct text-only training for video captioning for the first time and validate its effectiveness. We design in-context prompts to guide GPT-4 in generating diverse, high-quality domain-specific captions, which streamlines the construction of domain-specific corpora, minimizing reliance on paired data and associated costs.
- We propose a novel mixture of scale and shift architecture to address the distribution shift between synthetic and actual captions, which not only enhances the flexibility and performance of the model but also significantly reduces the number of parameters requiring updates.
- On three commonly used benchmarks, MoS$^2$ improves state-of-the-art few-shot video captioning methods by +13.6, +11.3 and +3.6 in CIDEr score. Remarkably, on MSRVTT dataset, MoS$^2$ with 10 real target-domain caption examples achieves 95.7% performance of the model with 130K real target-domain data.

## 2 RELATED WORKS

*Text-Only Captioning*. With the popularity of CLIP [35], recent researches [8, 19, 24, 31, 39, 41, 44, 56] attempt to improve image captioning by leveraging the aligned latent space. ZeroCap [41] aims to generate descriptions with the highest CLIP similarity to the image by combining CLIP with large language models. Despite the correlated CLIP text and visual latent space, the modality gap still hinders the generation of high-quality captions. DeCap [24] learns a text decoder on text-only data and projects visual embedding into CLIP text embedding space to reduce the modality gap, achieving excellent zero-shot image captioning performance. Nevertheless, these methods rely on pre-trained contrastive models, and the inherent discrepancy between generative and contrastive paradigms could potentially constrain performance. In response, our work seeks to enhance video captioning using pre-trained image captioning models. Furthermore, we utilize GPT-4 to generate domain-specific and high-quality corpora, aiming to bolster text-only captioning capabilities.

*Parameter-Efficient Learning*. With the increasing model sizes, fully fine-tuning all parameters becomes increasingly cumbersome. To reduce the computational burden, researchers explore alternative parameter-efficient learning methods that only train a few parameters [14, 15, 21, 25, 40, 55]. One pioneering approach [14] inserts lightweight adapters into transformer layers, achieving comparable performance with full fine-tuning. Prompt Tuning [21, 25, 58] appends a group of learnable prefix tokens to language models to guide downstream tasks, while keeping the entire model frozen. In this paper, we investigate representative parameter-efficient methods for video captioning and introduce a novel mixture of scale and shift architecture tuning methods, which improves the model's flexibility and performance while preserving a minimal number of trainable parameters.

*Mixture of Experts*. Mixture of Experts (MoE) models, characterized by their scalable capacity and enhanced performance, have gained prominence in the development of large language models [18]. These models leverage a trainable gating mechanism [38], which dynamically allocates inputs to specific experts, enabling sparse activation to augment capacity without significantly increasing computational demands. Nevertheless, achieving optimal load balance among the experts remains a challenge. To address this, [20] introduces a novel automatic load balancing strategy by incorporating a loss term that promotes more uniform activation of experts. Furthermore, [30] introduced an importance term to mitigate imbalances caused by the self-reinforcing effect of expert utilization. Building on these foundational advancements, our work introduces a novel architecture, the Mixture of Scale and Shift, which significantly enhances video captioning capabilities with while minimizing the increase in trainable parameters.

## 3 METHOD

In this section, we present our refined text-only video captioning pipeline. In Section 3.1, we introduce the text-only captioning setting. Subsequently, Section 3.2 details our innovative parameter-efficient tuning method, which allows the model to dynamically

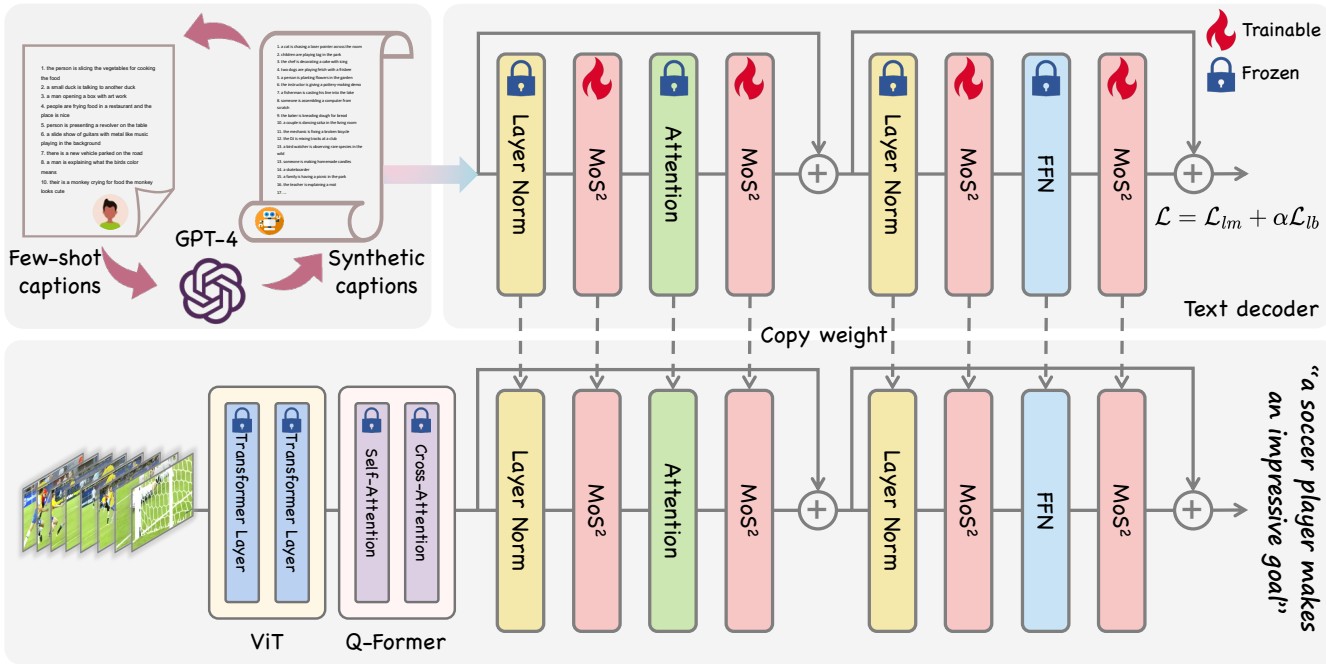

**Figure 1: Overview of our text-only video captioning pipeline. We develop a three-stage pipeline for text-only video captioning. Initially, using few-shot caption samples, we employ GPT-4 to generate in-domain captions. Subsequently, we enhance the text decoder through parameter-efficient fine-tuning, integrating a Mixture of Scale and Shift Expert layer (MoS²). Finally, the refined text decoder is combined with a visual encoder to facilitate video captioning.**

adjust its parameters in response to varying inputs. Section 3.3 discusses the training loss, while Section 3.4 describes the inference pipeline for video captioning. The overall pipeline is illustrated in Figure 1.

## 3.1 Text-Only Captioning Setting

***Text-Only Video Captioning Pipeline***. The development of visual captioning systems typically commences with the curation of a paired dataset. Unfortunately, the manual annotation of such datasets is both time-consuming and laborious. Curating video captioning datasets is even more arduous than image captioning datasets, as annotators must invest additional time to comprehend the dynamic and complex content of videos.

Recent research attempts text-only captioning, aiming to reduce the reliance on paired datasets. Text-only captioning seeks to enhance caption performance in specific domains using solely textual data. This task offers two configurations based on text data availability: full-text and few-shot. In the full-text scenario, all relevant textual data from the target domain are employed to optimize the text decoder, thereby maximizing performance within that domain. Conversely, the few-shot setting, characterized by limited text availability, does not support comprehensive model fine-tuning. Previous text-only captioning methods mainly focus on the full-text configuration. Nonetheless, full-text configuration

is usually impractical, primarily because caption datasets are typically manually annotated, which is both labor-intensive and time-consuming. Consequently, producing high-quality in-domain captions for target domains remains a challenging endeavor.

In response, our study explores an alternative setting, better suited for real-world applications. We aim to enhance captioning performance by leveraging minimal text samples from the target domain. We attempt to leverage LLMs' instruction-following and contextual reasoning capabilities to produce captions that are not only high in quality but also diverse and well-aligned with the target domain. We employ LLMs to exploit their instruction-following and contextual reasoning capabilities, aiming to produce captions that are not only high-quality but also closely aligned with the target domain. Subsequently, we employ the synthesized data to fine-tune the text decoder, enhancing the model's captioning performance within on the target domain.

***In-Domain Text Synthesis***. Inspired by the remarkable instruction-following capability of large language models (LLMs), we propose leveraging ChatGPT or language GPT-4 [1] to synthesize in-domain captions, thereby reducing annotation costs. Specifically, we randomly select 10 captions from the downstream dataset and construct a "few-shot" instruction-following prompt, as illustrated in Table 1.

 

---

**System**: You are an expert in video caption annotation.
**Human**: Your task is to generate diverse and plausible captions that are consistent with the following sample annotations: {five sample captions}.
**Assistant**: {another five captions}.
**Human**: Your generated captions are plausible. Please generate another 100 diverse and plausible captions that are consistent with the sample annotations. Your captions must cover diverse and plausible content that could be found in YouTube videos. Do not include any explanations, just provide the 100 generated captions.

**Table 1: Prompt for in-domain text synthesis.**

This prompt encourages the LLMs to generate captions that align with the provided sample captions. However, the text synthesized in this way has a high percentage of duplicates. To mitigate this, we incorporate the generated text back into the prompt. Specifically, we randomly sample captions from both the selected downstream and synthesized captions and then prompt ChatGPT or GPT-4 [1] with the novel prompt. This self-instructed technique injects randomness into the prompt, effectively reducing text repetition. In preliminary experiments, we compare the performance of Chat-GPT and GPT-4 [1] and find that GPT-4 consistently provides higher quality captions, consistent with [29]. Our methodology enables the synthesis of in-domain captions without the need to access visual data.

## 3.2 Mixture of Scale and Shift Experts

TTo match the distribution of a target dataset, SSF [26] incorporates scale and shift factors to modulate deep features extracted by a pre-trained model. Specifically, modifies an input token $x \in \mathbb{R}^d$ by scaling and shifting as follows:

$$z = x \odot \gamma + \beta, \tag{1}$$

where $\gamma \in \mathbb{R}^d$ and $\beta \in \mathbb{R}^d$ represent learnable affine transform parameters. $\odot$ denotes the dot product, and $z$ signifies the adjusted token. However, their scale and shift factors are input-independent, potentially constraining the model's expressive capacity. To address this limitation, we propose an adaptive learning mechanism for scale and shift parameters that tailors to the specific inputs:

$$z = x \odot \gamma_x + \beta_x, \tag{2}$$

where $\gamma_x$ and $\beta_x$ are input-dependent scale and shift factors, allowing for a more flexible and responsive model adaptation.

A straightforward method for determining input-dependent scale and shift factors involves employing an MLP to estimate these factors. However, such an approach introduces a significant number of learnable parameters, which can lead to unstable training and suboptimal performance, especially in scenarios with limited training data. Moreover, there is a distribution shift between synthetic and actual captions, further increasing learning difficulty. To mitigate these challenges, we propose a novel approach that constructs

a sparse Mixture of Scale and Shift expert model (MoS$^2$). We define a collection of discrete scale and shift experts, coupled with a lightweight routing network tasked with predicting a probability distribution among these experts. Subsequently, the top-k experts with the highest probabilities are aggregated to formulate the input-dependent scale and shift factors. The MoS$^2$ architecture aims to reduce the number of parameters to be updated and simplify the estimation of scale and shift factors. As opposed to SSF [26], our MoS$^2$ model can adaptively adjust to varying inputs and enhance the model's capacity, thereby improving performance in downstream tasks. The architecture of MoS$^2$ is depicted in Fig. 2.

The mixture of scale components comprises a routing network, denoted as $G$, in conjunction with $n$ scale experts, represented as $\Gamma = \{\gamma_1, \gamma_2, \ldots, \gamma_n\}, \gamma_i \in \mathbb{R}^d$. We derived the input-dependent scale factor $\gamma_x$ from the weighted summation of adaptively selected scale experts. Formally, for the input token $x$, the routing network generates an $n$-dimensional probability distribution $g(x)$ across the scale experts:

$$g(x) = \text{Softmax}\left(x \otimes W_g + \epsilon\right), \tag{3}$$

where $W_g \in \mathbb{R}^{d \times n}$ signifies the trainable weight of the router, $\otimes$ denotes matrix multiplication, and $\epsilon$ refers to Gumbel noise [17], facilitating weighted sampling from a distribution. It is flexible to choose the appropriate routing network and we choose a single linear layer for simplicity. Subsequently, we identify the top-$k$ scale experts with the highest probabilities and normalize the probability distribution as follows:

$$g_n(x) = \text{Norm}\left(\text{Top}\left(g(x), k\right)\right), \tag{4}$$

where $g_n(x)$ denotes the normalized distribution, $k$ signifies the number of experts chosen per token, and the Norm function adjusts the output by dividing each element by their cumulative sum. The input-adaptive scale factor is then computed via a weighted summation of the selected experts:

$$\gamma_x = \sum_{i=1}^{k} g_n(x)_i \odot \gamma_i. \tag{5}$$

In a parallel manner, we define a collection of shift experts, $B = \{\beta_1, \beta_2, \ldots, \beta_n\}$, and estimate their probability distribution $g'(x)$ via a shift router, described as:

$$g'(x) = \text{Softmax}\left(x \otimes W'_g + \epsilon\right), \tag{6}$$

where $W'_g$ denotes the trainable weight of the shift router. Subsequently, we identify top-$k$ shift experts with the highest probabilities and normalize the probability distribution:

$$g'_n(x) = \text{Norm}\left(\text{Top}\left(g'(x), k\right)\right). \tag{7}$$

The input-dependent shift vector is derived through a weighted aggregation of these selected shift experts:

$$\beta_x = \sum_{i=1}^{k} g'(x)_i \odot \beta_i. \tag{8}$$

Upon determining the input-dependent scale, $\gamma_x$, and shift, $\beta_x$, we proceed to adjust the hidden representations accordingly as prescribed in Eq. (2).

It is noteworthy that our approach could converge to SSF [26] if the router matrices $W_g$ and $W'_g$ are optimized to exhibit repetitive

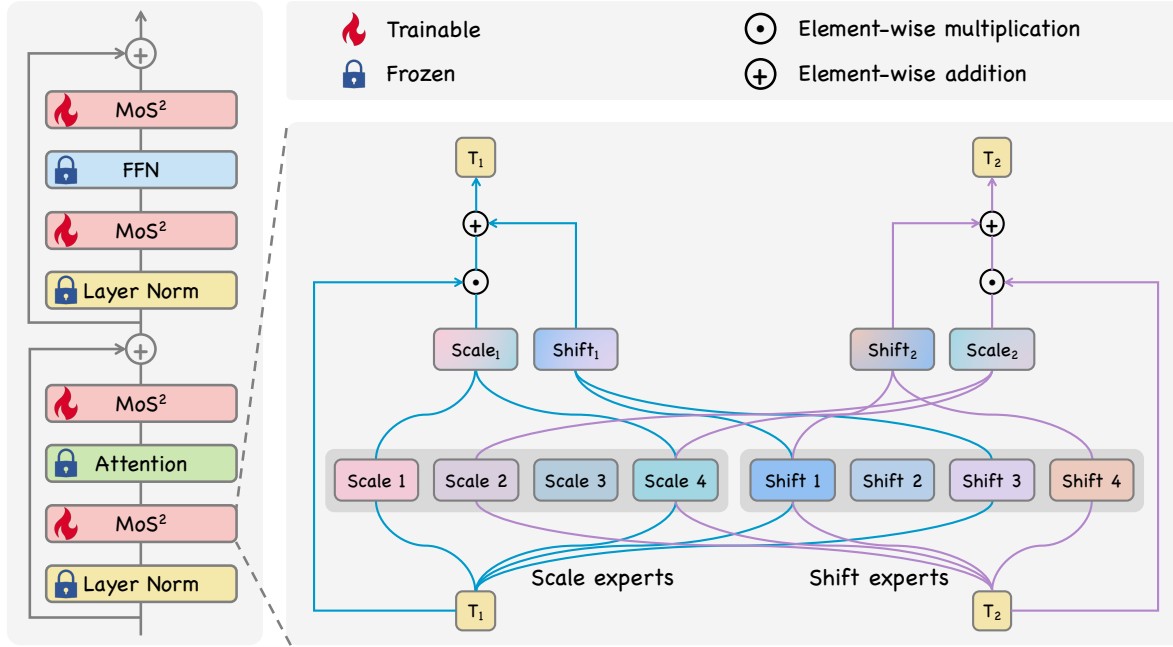

**Figure 2: Overview of Mixture of Scale and Shift expert model (MoS²). For each individual token, top-$k$ scale and shift experts are selected to construct input-dependent scale and shift factor. The factors are subsequently applied to themselves, enhancing the model's responsiveness to varying input conditions.**

column patterns. We allocate distinct parameters to the routers of scale and shift to the representation capacity of the model. Moreover, by increasing $n$ while maintaining $k$ constant, we are able to enhance the adaptability of the model without significantly affecting its computational demand. It operates independently per token and is orthogonal to the attention mechanism. During adaptation, we freeze the pre-trained weights, with updates applied exclusively to the MoS² parameters. The selective adaptability of MoS² enables the model to dynamically adjust its parameters in response to specific inputs, substantially improving the adaptability for downstream applications.

***Load Balancing Loss.*** Our MoS², characterized as a lightweight Mixture of Expert (MoE) architecture, confronts a load-balancing challenge in the absence of regularization mechanisms. Specifically, unregularized routing networks are prone to allocating disproportionate weights to a small subset of experts, regardless of the input. This imbalance is self-reinforcing [30], with preferred experts receiving enhanced training, thereby increasing their likelihood of selection in subsequent routing decisions. To counteract this imbalance, we introduce a load-balancing loss function designed that penalizes the over-reliance on any single expert. The importance of an expert is quantified as the mean routing probability assigned to that expert across a batch. Our objective is to ensure a balanced utilization of experts by penalizing those with high-importance scores. For a batch consisting of $T$ tokens, represented as $X = \{x_1, x_2, \ldots, x_T\}$, the load balancing loss, $\mathcal{L}_{lb}$, is formulated

as follows:

$$\mathcal{L}_{lb} = n \sum_{i=1}^{n} \bar{p}_i \odot q_i,$$

$$\bar{p}_i = \frac{1}{T} \sum_{x \in X} p_i(x), \qquad (9)$$

$$q_i = \frac{1}{T} \sum_{x \in X} \mathbb{1}\{\arg \max p(x) = i\},$$

where $n$ denotes the number of experts, $\bar{p}_i$ represents the batch-averaged routing probability for expert $i$, $p(x) \in \mathbb{R}^n$ signifies the router probability for token $x$, and $q_i$ indicates the proportion of tokens dispatched to expert $i$, $\mathbb{1}$ being the indicator function. The load balancing loss ensures that the training process remains balanced, fostering an equitable distribution of learning across all experts. The final load balance loss is computed by averaging across all mixture of scale and shift modules, which we omit it for simplicity. Through the load balancing loss, the training process is calibrated to ensure a balanced, equitable distribution of learning opportunities across all experts, thereby mitigating the risk of over-reliance on a limited subset of the available experts.

## 3.3 Text-Only Training

In light of the substantial costs and effort required for manually annotating paired datasets, our study aims to enhance video captioning through an innovative approach that relies solely on text data. We incorporated our designed MoS² layer into the text decoder $h$, positioned after layer normalization, multi-head attention, and pointwise feed-forward networks. The MoS² layer is fine-tuned

Table 2: Comparison with state-of-the-art methods on MSR-VTT [49] test split. Bold and underline indicate the best and second best performance respectively. The terms "All", "Full-text", and "Text" refer to the use of all data from the target domain, all text data, and partial text data, respectively.

| Method | All | Full-text | Text | B4 | M | R | C |
|---|---|---|---|---|---|---|---|
| **Zero-shot** | | | | | | | |
| Yue Zhao et al. [60] | ✗ | ✗ | ✗ | - | - | - | **48.2** |
| PaLM2-VAdapter [47] | ✗ | ✗ | ✗ | - | - | - | 47.7 |
| InternVideo2 [46] | ✗ | ✗ | ✗ | - | - | - | 43.5 |
| BLIP-2 [22] | ✗ | ✗ | ✗ | 29.5 | 25 | 53.4 | 48.0 |
| **Few-shot** | | | | | | | |
| Open-Flamingo-3B [3] | ✗ | ✗ | ✓ | 24.8 | 24.3 | 51.1 | 38.2 |
| Open-Flamingo-9B [3] | ✗ | ✗ | ✓ | 30.1 | 25.4 | 54.9 | 47.2 |
| MoS$^2$ (ours) | ✗ | ✗ | ✓ | **40.3** | **28.2** | **60.4** | **58.5** |
| **Full-text** | | | | | | | |
| CLIPRe [39] | ✗ | ✓ | ✓ | 10.2 | 18.8 | - | 19.9 |
| DeCap [24] | ✗ | ✓ | ✓ | 23.1 | 23.6 | - | 34.8 |
| MoS$^2$ (ours) | ✗ | ✓ | ✓ | **44.5** | **29.9** | **62.7** | **61.1** |
| **Fine-tuning** | | | | | | | |
| MV-GPT [37] | ✓ | ✓ | ✓ | 48.9 | 38.7 | 64 | 60 |
| Vid2Seq [51] | ✓ | ✓ | ✓ | - | 30.8 | - | 64.6 |
| HiTeA [53] | ✓ | ✓ | ✓ | - | - | - | 65.1 |

Table 3: Comparison with state-of-the-art methods on MSVD [5] test split. Bold and underline indicate the best and second best performance respectively. The terms "All", "Full-text", and "Text" refer to the use of all data from the target domain, all text data, and partial text data, respectively.

| Method | All | Full-text | Text | B4 | M | R | C |
|---|---|---|---|---|---|---|---|
| **Zero-shot** | | | | | | | |
| Uni-Perceiver [63] | ✗ | ✗ | ✗ | 20.3 | 25.8 | 52.1 | 45.7 |
| InternVideo2 [46] | ✗ | ✗ | ✗ | - | - | - | **93.1** |
| BLIP-2 [22] | ✗ | ✗ | ✗ | **38.9** | **35.0** | **68.5** | 87.1 |
| **Few-shot** | | | | | | | |
| Open-Flamingo-3B [3] | ✗ | ✗ | ✓ | 61.0 | 39.5 | 75.7 | 122.5 |
| Open-Flamingo-9B [3] | ✗ | ✗ | ✓ | 65.7 | 41.9 | 78.8 | 128.9 |
| MoS$^2$ (ours) | ✗ | ✗ | ✓ | **67.5** | **44.8** | **81.2** | **142.5** |
| **Full-text** | | | | | | | |
| MoS$^2$ (ours) | ✗ | ✓ | ✓ | 65.2 | 44.6 | 81.1 | 143.4 |
| **Fine-tuning** | | | | | | | |
| Uni-Perceiver [63] | ✓ | ✓ | ✓ | 61.5 | 42.3 | 79.0 | 131.0 |
| Vid2Seq [51] | ✓ | ✓ | ✓ | - | - | - | 146.2 |
| HiTeA [53] | ✓ | ✓ | ✓ | - | - | - | 146.9 |

on our synthesized in-domain captions. Specifically, we employ casual language modeling [36] to train MoS$^2$, which involves predicting the current text token conditioned on the preceding tokens. Formally, we minimize the following loss:

$$\mathcal{L}_{lm} = -\frac{1}{|t|} \sum_{i=1}^{|t|} \log h_\rho \left( t_i \mid t_{<i} \right), \quad (10)$$

where $\rho$ represents the parameters of the MoS$^2$ layer. The overall training loss incorporates both the causal language modeling loss, $\mathcal{L}_c$, and the load balancing term for MoS$^2$, $\mathcal{L}_b$, is expressed as:

$$\mathcal{L} = \mathcal{L}_{lm} + \alpha \mathcal{L}_{lb}, \quad (11)$$

with $\alpha$ representing the coefficient for the load balancing term. Our methodology, leveraging text-only training data, significantly reduces reliance on paired datasets. The text-only training paradigm, coupled with the incorporation of the MoS$^2$ layer, offers a promising avenue for advancing video captioning capabilities, without the necessity for extensive manual data annotation. After training, we can combine the visual encoder with the fine-tuned text decoder for video captioning.

## 3.4 Video Captioning Inference

We integrate our text-only training pipeline and the MoS$^2$ architecture into BLIP-2 [22] model for video captioning. BLIP-2 [22] connects a frozen visual encoder and a large language model by a lightweight perceiver-based [16] transformer, Q-Former, which facilitates the integration of image encoders with large language models, thereby bootstrapping vision-to-language generative modeling. For video inputs containing multiple frames, we simply concatenate the frame-level visual representations before processing with the Q-Former.

## 4 EXPERIMENTS

### 4.1 Experimental Setup

We start from BLIP-2 [22], a pre-trained image captioning model equipped with ViT-G from EVA-CLIP [9] and OPT-2.7B [59]. Further improvement can be achieved by augmenting the model with a more powerful visual encoder and textual decoder. We evaluate our approach on three popular video captioning benchmarks, including MSRVTT [49], MSVD [5], and VATEX [45]. We evaluate the quality of generated captions using four standard captioning evaluation metrics, including BLEU-4 (B4) [33], METEOR (M) [4], ROUGE-L (R) [27], and CIDEr (C) [42]. Additional experimental details are provided in the appendix.

### 4.2 Main Results

Table 2, Table 3 and Table 4 presents the video captioning results in both few-shot and full-text settings. Our method significantly surpasses the previous state-of-the-art in the few-shot setting, demonstrated by CIDEr score enhancements of +13.6, +11.3, and +3.6 on the MSVD, MSRVTT, and VATEX datasets, respectively. Additionally, compared to the zero-shot outcomes of BLIP-2, our approach leveraging few-shot in-domain text synthesis shows remarkable improvements, with increases in CIDEr scores of +55.4, +10.5, and +10.4 across these datasets. Notably, our few-shot result outperforms the fine-tuning result of Uni-Perceiver [63] on the MSVD dataset. Specifically, we attained a CIDEr score of 142.5, surpassing the score of 131.0 achieved by Uni-Perceiver [63], which is pre-trained on extensive image-text and video-text pairs. These results highlight the effectiveness of our novel few-shot in-domain text synthesis and a mixture of scale and shift tuning methods. Additionally, the minor discrepancies observed between our few-shot and full-text results suggest that our text generation technique successfully captures the critical elements of the original datasets, underscoring the robustness of our methods. In the full-text setting, our approach outperforms the state-of-the-art method DeCap [24], by a margin of +26.3 on MSRVTT, recording a CIDEr score of 61.1

**Table 4: Comparison with state-of-the-art methods on VA-TEX [45] public test split. Bold and underline indicate the best and second best performance respectively. The terms "All", "Full-text", and "Text" refer to the use of all data from the target domain, all text data, and partial text data, respectively.**

| Method | Video | Full-text | Text | B4 | M | R | C |
|---|---|---|---|---|---|---|---|
| **Zero-shot** | | | | | | | |
| ChatBridge [61] | ✗ | ✗ | ✗ | - | - | - | 48.9 |
| OneLLM [12] | ✗ | ✗ | ✗ | - | - | - | 43.8 |
| EffShortViViT [32] | ✗ | ✗ | ✗ | - | - | - | 43.6 |
| PaLM2-VAdapter [47] | ✗ | ✗ | ✗ | - | - | - | **53.0** |
| InternVideo2 [46] | ✗ | ✗ | ✗ | - | - | - | 49.2 |
| BLIP-2 [22] | ✗ | ✗ | ✗ | 21.7 | 19.9 | 45.8 | 39.9 |
| **Few-shot** | | | | | | | |
| Open-Flamingo-9B [3] | ✗ | ✗ | ✓ | 25.4 | 20.2 | 46.8 | 41.8 |
| Flamingo-3B [2] | ✗ | ✗ | ✓ | - | - | - | 40.1 |
| Flamingo-9B [2] | ✗ | ✗ | ✓ | - | - | - | 39.5 |
| Flamingo-80B [2] | ✗ | ✗ | ✓ | - | - | - | 46.7 |
| MoS$^2$ (ours) | ✗ | ✗ | ✓ | 28.7 | 21.9 | 46.7 | 50.3 |
| **Full-text** | | | | | | | |
| DeCap [24] | ✗ | ✓ | ✓ | 21.3 | 20.7 | - | 43.1 |
| MoS$^2$ (ours) | ✗ | ✓ | ✓ | 28.6 | 22.6 | 47.4 | 53.1 |
| **Fine-tuning** | | | | | | | |
| PaLI-3 [7] | ✓ | ✓ | ✓ | - | - | - | 66.9 |
| SwinBERT [28] | ✓ | ✓ | ✓ | 38.7 | 26.2 | 53.2 | 73.0 |
| VideoCoCa [50] | ✓ | ✓ | ✓ | 39.7 | - | 54.5 | 77.8 |

**Table 5: Comparison with state-of-the-art parameter-efficiency learning methods.**

| Method | Params | B4 | M | R | C | Mean |
|---|---|---|---|---|---|---|
| SSF [26] | **0.2M** | 39.2 | 27.4 | 58.9 | 56.8 | 45.6 |
| BitFit [55] | 0.9M | 38.5 | 27.2 | 59.0 | 55.9 | 45.1 |
| LLaMA-Adapter [58] | 1.3M | 39.1 | 27.5 | 59.6 | 57.0 | 45.8 |
| LoRA [15] | 13.1M | 39.1 | 27.3 | 59.1 | 56.8 | 45.6 |
| Adapter [14] | 52.6M | 39.7 | 27.7 | 59.7 | 57.8 | 46.2 |
| Full fine-tuning | 2,651.6M | 38.9 | 27.8 | 59.5 | 57.3 | 45.9 |
| MoS$^2$ (ours) | 10.5M | **40.3** | **28.2** | **60.4** | **58.5** | **46.8** |

against 34.8. These findings suggest that our parameter-efficient tuning approach adeptly adapts pre-trained image captioning models for video captioning tasks, effectively narrowing the performance gap between zero-shot and fine-tuning results.

## 4.3 Ablation Study

We conduct comprehensive ablation analysis on the MSR-VTT [49] dataset with our synthesized corpus. We use mean of BLEU-4 [33], METEOR [4], ROUGE-L [27], and CIDEr [42] as the primary evaluation metric for robustness.

***Parameter Efficiency.*** We conduct a comparative analysis of the state-of-the-art parameter-efficient tuning methods, as detailed in Table 5. Our proposed tuning method, MoS$^2$, demonstrates superior performance across all metrics while maintaining competitive trainable parameters, at a reduced parameter count of 10.5M. This denotes a notable advancement in learning efficiency, achieving a delicate balance between parameter count and performance. The

**Table 6: Ablation on number of experts $n$.**

| $n$ | Params | B4 | M | R | C | Mean |
|---|---|---|---|---|---|---|
| 4 | **1.3M** | 39.1 | 27.7 | 59.4 | 57.2 | 45.8 |
| 8 | 2.6M | 39.1 | 27.8 | 59.5 | 57.3 | 45.9 |
| 16 | 5.2M | **39.6** | **28.0** | 59.8 | 57.8 | 46.3 |
| 32 | 10.5M | **39.6** | **28.0** | **60.0** | **57.9** | **46.4** |

**Table 7: Ablation on top-$k$.**

| $k$ | B4 | M | R | C | Mean |
|---|---|---|---|---|---|
| 2 | **39.6** | **28.0** | **60.0** | **57.9** | **46.4** |
| 4 | 39.5 | 27.8 | 59.5 | 57.5 | 46.1 |
| 8 | 39.4 | 27.9 | 59.5 | 57.8 | 46.1 |
| 16 | **39.6** | 27.8 | 59.6 | 57.7 | 46.2 |
| 32 | **39.6** | 27.8 | 59.6 | 57.8 | 46.2 |

SSF [26] method introduces input-independent scale and shift factors, requiring a minimal number of trainable parameters (0.2M), and demonstrates significant parameter efficiency by delivering commendable performance. Conversely, BitFit [55], which adjusts only the bias term, fails to achieve a performance increase. The LLMaM-Adapter [58] method, which prepends learnable adaptation prompts to word tokens, exhibits enhanced performance with a slightly increased parameter count, underscoring the efficacy of prompts for fine-tuning Transformer models. The above methods are characterized by a limited number of trainable parameters, wherein only trainable vectors are employed. LoRA introduces trainable rank decomposition matrices into each layer of the Transformer architecture, resulting in a considerable increase in the parameter count. Adapter [14] utilizes 52.6M parameters and achieves the second-best result. Full fine-tuning, with the most trainable parameters, does not achieve the best performance, implying that no need to update so many parameters. Similarly, the Adapter incorporates MLP, further increasing the total number of trainable parameters to 52.6 million, achieving the second-best performance. Notably, full fine-tuning, which utilizes the highest number of trainable parameters, does not yield the best results, suggesting that updating an extensive array of parameters is not necessary for optimal performance.

***Number of Experts.*** The quantity of expert models plays a crucial role in MoE-based models. We attempt to increase the number of scale and shift expert models, as shown in Table 6. Our experimental findings indicate a direct correlation between an increase in the number of expert models and enhanced model performance. Nonetheless, considering the parameter efficiency, we ultimately select 32 expert models, opting not to expand further.

***Top-$k$ Routing.*** Presented in Table 7, we investigate the effect of varying the number of expert models ($k$) selected by each token. Our findings indicate that increasing $k$ does not improve performance. Therefore, we set $k = 2$ for subsequent analyses.

***Load Balance Loss Coefficient.*** As load balancing significantly influences the performance of MoE models, we incorporate a load balance loss term into our method. As demonstrated in Table 8, an appropriate load balance loss coefficient, specifically set at $\alpha = 0.1$, notably improves performance by +3.3. This finding highlights the

**Table 8: Ablation on load balance loss coefficient $\alpha$.**

| $\alpha$ | B4 | M | R | C | Mean |
|---|---|---|---|---|---|
| 1 | 39.6 | 28.0 | 60.0 | 57.9 | 46.4 |
| 0.2 | 39.6 | **28.1** | 60.3 | 57.8 | 46.5 |
| 0.1 | **40.0** | **28.1** | 60.4 | **58.3** | **46.7** |
| 0.05 | 39.8 | 28.0 | **60.5** | 58.0 | 46.6 |
| 0 | 36.3 | 27.2 | 57.8 | 52.3 | 43.4 |

**Table 9: Ablation on mixture of scale and shift.**

| Scale | Shift | Params | B4 | M | R | C | Mean |
|---|---|---|---|---|---|---|---|
| ✓ |  | **5.2M** | 34.7 | 26.5 | 56.1 | 50.3 | 41.9 |
|  | ✓ | **5.2M** | 40.1 | 28.1 | 60.1 | 58.4 | 46.7 |
| ✓ | ✓ | 10.5M | **40.3** | **28.2** | **60.4** | **58.5** | **46.9** |

**Table 10: Ablation on number of MoS$^2$ layers.**

| Type (#Layers) | Params | B4 | M | R | C | Mean |
|---|---|---|---|---|---|---|
| Interleaved (4) | **5.2M** | 40.1 | 28.1 | 60.2 | 57.9 | 46.6 |
| Interleaved (8) | 10.5M | 40.0 | 28.1 | 60.4 | 58.3 | 46.7 |
| Interleaved (16) | 21.0M | 39.5 | 28.0 | **60.5** | 57.7 | 46.4 |
| First (8) | 10.5M | 37.5 | 27.5 | 58.9 | 54.0 | 44.5 |
| Middle (4) | **5.2M** | **40.3** | 28.1 | 60.2 | **58.7** | 46.8 |
| Middle (8) | 10.5M | **40.3** | **28.2** | 60.4 | 58.5 | **46.9** |
| Middle (16) | 21.0M | 39.2 | 28.1 | 60.3 | 57.2 | 46.2 |
| Last (8) | 10.5M | 38.4 | 27.5 | 59.6 | 56.3 | 45.5 |

essential role of the load balance loss term in boosting the efficacy of MoE models.

***Different Components.*** We conduct ablation experiments on the two expert models, specifically analyzing the effects of removing the mixture of scale or the mixture of shift components. As indicated in Table 9, omitting the mixture of shift results in a significant performance decrease of -5, whereas eliminating the mixture of scale leads to a marginal reduction of -0.2. We hypothesize that the more pronounced effect of omitting the mixture of shift could be attributed to its incorporation as residuals to the hidden states, which likely stabilizes the gradient and simplifies the training process. In contrast, the mixture of scale directly multiplies the hidden states, potentially introducing greater variability in model training.

***Insertion Layers.*** We investigate the distribution of MoS$_2$ layers within Transformer models to determine their optimal placement. Our findings, presented in Table 10, indicate that incorporating MoS$_2$ into the middle eight layers of the Transformer significantly enhances performance relative to other configurations such as the initial or final eight layers, or an interleaved arrangement. Additionally, we observed that an increase in the total number of layers does not consistently improve performance; rather, the most effective enhancement is achieved through the middle eight layers.

***Insertion Locations.*** We analyze the placement of components within the Transformer block. As demonstrated in Table 11, removing MoS$^2$ subsequent to layer normalization substantially degrades

**Table 11: Ablation on MoS$^2$ location within Transformer block.**

| Location | Params | B4 | M | R | C | Mean |
|---|---|---|---|---|---|---|
| w/o norm | **5.2M** | 39.0 | 28.0 | 59.8 | 57.5 | 46.1 |
| w/o attn | 7.9M | 40.1 | 28.0 | 60.3 | **58.3** | **46.7** |
| w/o FFN | 7.9M | **40.2** | 28.0 | 60.2 | 57.7 | 46.5 |
| MoS$^2$ (ours) | 10.5M | 40.0 | **28.1** | **60.4** | 58.3 | **46.7** |

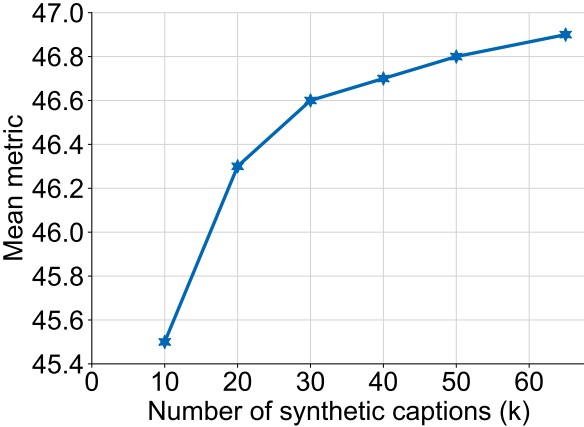

**Figure 3: Performance of MoS$^2$ under different training data scales.**

performance, more so than its removal after the feed-forward network (FFN). Conversely, eliminating MoS$^2$ subsequent to the attention layers has a negligible impact on performance; nevertheless, we maintain MoS$^2$ in this position by default.

***Number of Text.*** We evaluate the impact of increasing the number of synthetic text on model performance, as depicted in Fig. 3. Our results indicate a consistent enhancement in model efficacy correlating with increased text quantities, suggesting potential for further gains. This trend demonstrates that the synthetic texts generated by our approach effectively encapsulate critical aspects of the target domain knowledge.

## 5 CONCLUSION

This work presents a refined text-only training pipeline that significantly enhances the usability and performance of domain-specific captioning. By leveraging GPT-4 to create high-quality, domain-specific captions, we successfully reduce dependency on extensive paired data, thereby reducing associated annotation costs and labor. We also introduce the Mixture of Scale and Shift Experts (MoS$^2$) architecture, which effectively mitigates distribution shifts between synthetic and real captions, thereby enhancing model flexibility. MoS$^2$ outperforms state-of-the-art methods in both few-shot and full-text video captioning across three established benchmarks. Remarkably, MoS$^2$ achieves near full-text model performance with only minimal real training examples (10-shot), demonstrating the viability of using large language models (LLMs) to generate training data and underscoring the potential of parameter-efficient learning techniques in domain-specific applications.

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
