# OpenReview forum: "MoS$^2$: Mixture of Scale and Shift Experts for Text-Only Video Captioning"
_acmmm.org/ACMMM/2024/Conference — MM2024 Poster_

### Official Review · Reviewer_eVwV · 2024-05-22

**Rating:** 2
**Confidence:** 4

**Summary:**

Video captioning is a complex task that generally needs video-text paired data for training, but manually creating coherent text descriptions for videos is a labor-intensive and time-consuming process. The paper proposes a video captioning pipeline that begins with data expansion using GPT-4, designs a PEFT method called MoS2 for text-only training, and does video captioning inference. However, there are still some issues in the paper that need to be addressed.

**Strengths:**

1. The pipeline proposed in this paper is particularly clear and easy to follow, which makes it accessible even to those who may not be experts in the field. This straightforward approach ensures that researchers can understand and apply the methods discussed.
2.	The paper writing is commendable for its clarity and straightforwardness. The authors use simple and direct language, making complex concepts more understandable.
3. The paper conducts sufficient experiments to verify the effectiveness of the proposed method for video captioning.

**Limitations:**

1. The proposed PEFT method, MoS², is used to fine-tune a strong text decoder. I suggest comparing it directly with common PEFT methods like LoRA or adapters across additional NLP tasks, such as text generation.
2. I question the relevance of this paper to video captioning. In fact, BLIP2 is inherently suited for image captioning, not video captioning. I recommend testing it on image captioning tasks. Moreover, in the original BLIP2, a 224x224 resolution image corresponds to one cls token and 256 patch tokens, which interact with 32 learnable tokens. However, if tokens from different frames are merely stacked together in video captioning, it could neglect essential frame-wise temporal modeling.
3. Line 157 in the paper "We conduct text-only training for video captioning for the first time and validate its effectiveness" is not precise. Text-only methods such as DeCap[1] and MultiCap[2] have previously conducted experiments for video captioning on the MSR-VTT and VATEX datasets,  indicating that this approach is not novel.
4. On line 382, "TTo match" should be corrected to "To match" to maintain the professional quality of the manuscript.

Reference:

 [1] Li, Wei, et al. "DeCap: Decoding CLIP Latents for Zero-Shot Captioning via Text-Only Training." The Eleventh International Conference on Learning Representations. 2022.

   [2] Yang, Bang, et al. "MultiCapCLIP: Auto-Encoding Prompts for Zero-Shot Multilingual Visual Captioning." The 61st Annual Meeting Of The Association For Computational Linguistics. 2023.

**Suitability:**

3

---

### Official Review · Reviewer_89hb · 2024-05-24

**Rating:** 5
**Confidence:** 4

**Summary:**

The paper introduces a novel method for text-only video captioning, named MoS2. The authors leverage a large language model to create high-quality and high-diversity captions in the target domain for training. The proposed MoS2, a Mixture of Scale and Shift experts, allows a large-scale pre-training model for parameter-efficient fine-tuning. It significantly outperforms existing methods in three video captioning datasets.

**Strengths:**

1.	The proposed MoS2 effectively addresses the gap between generated data and target data. It enhances the training efficiency of LMM in domain transfer learning.
2.	The paper is well-written and clearly illustrated.
3.	The ablation study in the paper is comprehensive.

**Limitations:**

1.	The authors could consider adding some visualization results (e.g. case study, caption distribution before and after MoS2 adjustment).
2.	The authors should correct the typos and formatting errors in the paper.
         e.g.
         * line109 "[2, 54? ]"
         * line381 "TTo match"

**Suitability:**

3

---

### Official Review · Reviewer_ziHB · 2024-05-25

**Rating:** 4
**Confidence:** 2

**Summary:**

The author proposed a text-only training pipeline for video captioning that reduces the dependency on paired data. They also introduce the Mixture of Scale and Shift Experts to mitigate distribution shifts between synthetic and real captions.

**Strengths:**

1. The authors conduct text-only training for video captioning which not only is parameter-efficient but also minimizes reliance on paired data
2. The authors design a mixture of scale and shift architecture to address the distribution shift between synthetic and actual captions
3. The method achieves great performance on three established benchmarks
4. The motivation is clear and the idea is easy to follow

**Limitations:**

1. Poor presentation. Line 330 and line 333 seem to be duplicated. Line 110: captioning models [2, 54? ]
2. Could the synthetic data used by the authors also improve DeCap's performance?

**Suitability:**

2

---

### Meta-Review · Area_Chair_xiw7 · 2024-06-27

**Recommendation:** Accept (Poster)
**Confidence:** 5

**Metareview:**

The paper introduces a text-only training pipeline for video captioning, reducing dependency on paired data. After author rebuttal, most of the reviewers acknowledged that the motivation is clear and the paper includes sufficient experiments. However, there are typos and formatting errors, and some expressions are not precise. Overall, we would like to recommend Accept (Poster) of the paper.